# Exploring current and potential roles of informal healthcare providers in tuberculosis care in West Bengal, India: A qualitative content analysis

Poshan Thapa[1,2]*, Padmanesan Narasimhan[1], John J. Hall[1], Rohan Jayasuriya[1], Partha Sarathi Mukherjee[3], Dipesh Kr Das[3], Kristen Beek[1]

**1** School of Population Health, University of New South Wales, Sydney, Australia, **2** School of Population and Global Health, McGill University, Montreal, Canada, **3** Liver Foundation, Kolkata, West Bengal, India

* thapaposhan2009@gmail.com

## Abstract

India accounts for 27 percent of global Tuberculosis (TB) cases, the highest among the 30 high-burden countries. Despite growing evidence highlighting the significance and potential of Informal Healthcare Providers (IPs) in TB care, their role remains ambiguous in India's TB policies and programs, in contrast to the well-defined roles of the formal private sector. Considering such gaps, this study explores the perspectives of IPs (specifically untrained allopathic practitioners, UAPs) and National TB Elimination Program (NTEP)-affiliated personnel regarding IPs' current and potential roles in TB care. The study was conducted in West Bengal, India. We adopted a qualitative approach and conducted in-depth interviews with 23 IPs and 11 NTEP-affiliated personnel. The study data was analysed using a content analysis approach. The study's findings identified four current roles of IPs in TB care, two of which were corroborated by NTEP-affiliated personnel: 1) Passive case finding and referral and 2) Treatment supporter. As for potential roles, an alignment was observed between the two groups of providers for the majority of the roles (5/7 roles). However, both IPs and NTEP-affiliated personnel expressed reservations about assigning IPs the roles of 1) Clinical evaluation of people with TB and 2) Initiation of treatment for confirmed people with TB. The findings highlight the active involvement of IPs in various TB care roles, acknowledged by NTEP-affiliated personnel, and also demonstrate significant potential for their expanded engagement under the NTEP of India.

## Introduction

India accounts for 27 percent of the global burden of Tuberculosis (TB), with an estimated 2.8 million cases, the highest among the 30 high-TB burden countries [1]. A study examining the incidence and mortality of TB in India (between 1990 and 2019) reports a reduction in TB infections across all age groups. However, this progress is insufficient to meet the goals set out in India's National Strategic Plan (NSP) for TB

**Data availability statement:** All relevant data are within the paper and its Supporting Information files.

**Funding:** This work was supported in part by the Bill & Melinda Gates Foundation (Grant Number: INV-042531 to PT). Under the grant conditions of the Foundation, a Creative Commons Attribution 4.0 Generic License has already been assigned to the author-accepted manuscript version that might arise from this submission. The funders had no role in the study design, data collection and analysis, or the preparation of the manuscript.

**Competing interests:** The authors have declared that no competing interests exist.

Elimination (2017–2025) [2]. The NSP recommends strengthening engagement with the private sector, as approximately 50% of people with TB are managed outside the National TB Elimination Program (NTEP) [3,4]. In India, the private health sector ranges from small clinics to multispecialty hospitals and from informal healthcare providers (IPs) to highly qualified specialists. IPs are reported to provide a significant proportion of primary care, up to 70% in many settings, especially in rural and under-served communities [5–8]. Often referred to as village doctors or rural medical practitioners in India, IPs are providers who "typically operate outside the formal health system and lack accredited qualifications, frequently dispensing allopathic treatments such as antibiotics and injections without formal training" [9–11].

Despite the NSP's recommendation to enhance private sector involvement in TB care, IPs remain an under-prioritized group within India's NTEP. Nevertheless, there is accumulating evidence highlighting the significance and potential of IPs in TB care [9]. Several studies examining the health-seeking behaviours of people with TB have reported IPs as the initial or preferred providers in their care pathways, with some patients frequently visiting IPs before accessing government health facilities [12–14]. Additionally, research, including a study by our team involving 203 IPs, has documented the roles of IPs in managing TB by interacting with both presumptive and confirmed people with TB at the community level [15–17]. Moreover, various studies, along with a scoping review conducted by our team, have demonstrated positive TB care outcomes when IPs were engaged and supported, such as enhanced case referral and detection and improved treatment outcomes [10,18–20]. These findings suggest that the systematic inclusion of IPs could significantly benefit India's NTEP and aid in achieving the goals outlined in the NSP.

It is noteworthy that in several Indian states, practising as an IP is illegal and subject to penalties, adding complexity to their integration into health policies and programs, including TB care [21]. West Bengal (WB), the study site for this research, is unique as IPs are recognized by the state government, which issued an official order in November 2015 to train these providers [22,23].

The NSP of India recommends engaging all providers, public and private, formal and informal, within the framework of the public-private mix (PPM) in TB care. While there has been significant progress in engaging the formal private sector, the informal sector, despite being recognised as a stakeholder in PPM by NTEP, remains formally under-engaged in most states, with only a few exceptions like West Bengal [11,24,25]. In WB, efforts have recently begun to formally involve IPs in presumptive TB screening and referral functions. However, our previous research has identified multiple gaps in this engagement model, such as reliance on paper-based referral systems and inconsistent disbursement of referral incentives to IPs [11].

Furthermore, while clear policies and operational guidelines exist for engaging the formal private sector in TB care, there is a lack of corresponding clarity for IPs [24,25]. Therefore, given the ambiguity in India's TB policies and guidelines regarding IPs' roles and the variability in their engagement, ranging from limited to none across most states, it is crucial to thoroughly explore the roles currently undertaken by IPs

(either formally or informally) in TB care at the community level, as well as examine the potential roles IPs can undertake within the NTEP.

This study was conceived as part of broader multi-method research conducted in West Bengal, India, aimed at understanding the roles and engagement of IPs in TB care. Previous research from this project has been published [9–11,17,26], and this manuscript reports on the qualitative component of the project, specifically examining the roles of IPs in TB care.

Initially, a scoping review was conducted by our team to map the roles of IPs in TB care across Low and Middle-Income Countries (LMICs) [10]. The review found that IPs were involved in six distinct roles in TB care within three primary domains: prevention, detection, and treatment. These findings provided a foundation for this qualitative research, which seeks to further explore and deepen our understanding of IPs' roles at the field level in TB care. The scoping review categorized IPs into four groups based on their systems of practice: Traditional Health Practitioners, Untrained Allopathic Practitioners (UAPs), Traditional Birth Attendants, and Drug Sellers/Chemists [10]. This study, however, focuses exclusively on UAPs, as they were the least studied group in the scoping review. Concentrating on UAPs aligns with the broader multi-method research objectives. Notably, UAPs stand out for providing curative services within communities using allopathic medicine, including antibiotics, a practice with significant implications for TB case management [17,27,28]. Thus, the use of the term 'IPs' in this paper henceforth refers only to this subgroup of providers (UAPs).

This qualitative study also included the perspectives of NTEP-affiliated personnel, given their critical role in shaping policies and programs related to IPs. By incorporating insights from both IPs and NTEP-affiliated personnel, the study aimed to comprehensively explore the current roles of IPs in TB care and assess their potential for expanded involvement within the NTEP framework.

## Materials and methods

### Design

This study employed an exploratory qualitative design, appropriate for examining under-researched or complex phenomena from participants' perspectives. The analysis was guided by the principles of content analysis [29]. Content analysis was chosen as it allows for a structured and systematic examination of textual data, enabling the condensation of extensive data into meaningful concepts or categories [30]. This approach was particularly suitable for exploring the roles of IPs in TB care, as it facilitated the classification of roles based on functional categories informed by our previous scoping review [10].

The use of content analysis in this study was informed by our previous scoping review, which categorized IPs' roles in TB care according to the World Health Organization's (WHO) people-centered model of TB care [10,31]. This model outlines 17 standard roles across three domains: prevention, detection, and treatment. In the scoping review, six specific roles of IPs were identified within these domains: 1) health promotion and education, 2) active case finding, 3) passive case finding and referral, 4) laboratory sputum sample collection, 5) treatment supporter, and 6) prevention and detection of adverse events. These findings served as the foundation for developing a categorization matrix for the analysis in this study.

The WHO people-centered model was deemed comprehensive and appropriate for this study, as it encompasses a wide range of roles that can be undertaken by various healthcare practitioners throughout the continuum of TB care. This framework provided a systematic coding matrix to analyze the data. However, to ensure flexibility and rigour, we adopted an unconstrained approach during the coding process, remaining open to identifying any new roles emerging from the data. This approach resulted in the addition of two new role categories: 1) accompanying presumptive people with TB to a health facility and 2) contact tracing. These additional roles, along with the original six, are detailed in the supplementary file (S1 Text).

## Setting

This qualitative research was conducted as part of a broader multi-method study carried out in West Bengal, India, with the overarching goal of understanding the roles and engagement of IPs in TB care [9–11,17,26]. Before this qualitative study, a quantitative survey was undertaken to explore IPs' TB care practices [17]. IPs who participated in that study and provided consent formed the sampling frame for this qualitative research. The entire multi-method research project was conducted in collaboration with the Liver Foundation (LF), a local non-government organization with prior experience working with IPs. LF's established relationship with IPs was instrumental in building trust with the study participants and facilitated their enrolment.

The study was carried out in the Birbhum district of West Bengal, India, one of the working districts of LF. According to the 2023 India TB report, West Bengal reported 100,972 people with TB, accounting for only 77.7% of the expected cases [32]. Birbhum district, part of the Burdwan division, covers an area of 4,545 square kilometres. It has a population of 3,502,404, with a literacy rate of 70.9%. Agriculture is the predominant occupation, with 36% of the population belonging to scheduled tribes or castes. The per capita income (annual) in the district is 53,122 Indian rupees (approximately 680 USD) [33].

## Participants and sampling

This qualitative study was nested within a two-phase research project conducted in the Birbhum district. In Phase 1, a knowledge survey was carried out with a sampling frame of 589 IPs, developed in collaboration with the LF and the local IP association [26]. These IPs were located across all 19 administrative blocks of Birbhum, delivering care in community settings through private clinics and home-based visits. In Phase 2, 203 IPs from this initial frame participated in a second survey on TB care practices. Eligibility for the Phase 2 survey required that providers had managed at least one confirmed TB case in the preceding six months [17]. This criterion aligned with the objectives of this qualitative study, which aimed to explore the current and potential roles of IPs in TB care based on their recent experience.

During Phase 2, verbal consent for future re-contact was obtained from participants, allowing researchers to approach them for participation in subsequent research. For this qualitative study, the inclusion criteria for the first group of participants were: (a) participation in the Phase 2 survey and (b) willingness to take part in a follow-up interview. From the pool of 203 eligible IPs, 30 were purposively selected by two researchers (PT and KB) to ensure variation in age, education, and years of experience. Twenty-three agreed to participate, while seven declined due to time constraints, as they were busy with service delivery functions related to the COVID-19 pandemic.

Inclusion criteria for the second group of participants were: (a) NTEP-affiliated personnel working in the NTEP of India at the state, district, and community levels, thereby representing various health system tiers, and (b) willingness to participate in an interview. This group included personnel responsible for policy and program implementation at the state and district levels as well as Accredited Social Health Activists (ASHAs) who deliver TB care services at the community level. Throughout this study, the term "NTEP-affiliated personnel/an NTEP-affiliated participant" refers to individuals either working with or affiliated with the NTEP, including administrative and community-based roles.

Data saturation is a contested topic in qualitative research, with various guidelines suggesting different sample sizes needed to achieve saturation [34,35]. However, we did not aim for a specific sample size due to the exploratory nature of the study and the constraints imposed by the COVID-19 pandemic in India. Consequently, participation in the study was largely dependent on participants' availability, as both groups of participants were busy addressing the pandemic during the data collection period.

## Data collection tool and method

Our previous research work informed the interview guide [10,11,17,26], such as the probes on IPs' various roles in TB care. We pre-tested the study tool among four participants, but no significant changes were required. A copy of the study

tool is available as a supplementary file (S2 Text). Interviews with IPs began with icebreaker questions about their general experience working in the community. They were then asked to discuss their recent interactions with people with TB, followed by their experiences providing various TB care services within the community. The final section of the interview explored potential roles they would be willing to undertake as part of the NTEP. Similarly, interviews with NTEP-affiliated personnel started with icebreaker questions about their roles in TB care, followed by inquiries into their perspectives on the types of TB care services currently provided by IPs in the community. The discussion then shifted to potential roles IPs could play in NTEP.

Most of the interviews were conducted by a research assistant (RA), a male with a PhD in Health Sciences and over five years of experience working with IPs. The RA conducted the interviews in Bengali, the local language. The first author (PT), a male PhD candidate in Public Health with experience in qualitative research, conducted three interviews with NTEP-affiliated personnel who were comfortable speaking English. Data collection took place from October to December 2021, employing both phone and in-person interviews according to the participants' preferences. All interviews with IPs were conducted remotely by phone. For NTEP-affiliated personnel, four interviews were conducted in person, while the remaining seven were carried out via telephone. For in-person interviews, local COVID-19 guidelines were followed. Most interviews were recorded (n = 26), but notes were taken for participants (n = 8) who expressed discomfort with being recorded. The duration of the interviews varied between 30–40 minutes.

## Data analysis

The interviews were transcribed directly into English by the RA, who also participated in data collection, and the first author (PT). The data were analyzed using content analysis, a structured method that employs predefined coding categories based on prior research or existing theory. This approach was deemed appropriate as it allowed the condensation of interviews from both groups of providers into meaningful categories, providing a detailed description of roles from the participants' perspectives [36,37]. We followed three steps for content analysis as outlined by Elo and Kygnas [29].

**Preparation phase:** During the preparatory phase, two authors (PT and KB) independently read and re-read a sample of transcripts to become familiar with the data. They formulated specific guiding questions to frame their analysis, focusing on the roles expressed by both groups of providers, notable differences between them, and variations in current and potential roles.

**Organizing phase**: A categorization matrix of IP roles in TB care was established, informed by our prior scoping review [10]. The matrix followed an unconstrained approach, allowing for the identification of additional roles that emerged from the data. This method enhances the trustworthiness of findings by remaining open to new insights [38]. To ensure the matrix's relevance within the local context, India's Technical and Operational Guidelines for TB control were also consulted, though no new categories were identified [39]. The categorization matrix and its definitions are detailed in the supplementary file (S1 Text).

Using NVivo software version 12 (QSR International Pty Ltd., 2020), one author (PT) coded the transcripts based on the predefined matrix. Weekly meetings between PT and KB facilitated discussions on coding and the overall analysis process. Post-coding, PT and KB reviewed unassigned statements of relevance, integrating them into existing themes or forming two new categories: 'accompanying presumptive people with TB to a health facility' and 'contact tracing.

**Reporting findings:** We were interested in exploring IPs' current and potential roles in TB care as expressed by two groups of providers (informal versus formal). Therefore, we presented mapped categories defined as "IP roles" in tabular formats. The Consolidated Criteria for Reporting Qualitative Research (COREQ) 32-item checklist was followed for reporting consistency (S1 COREQ Checklist in S1 File) [40].

To protect the confidentiality of NTEP-affiliated personnel, we only indicate that the presented quotes belong to NTEP-affiliated personnel without providing further details, as any additional information could reveal their identity, considering each provider's unique role and responsibilities in the NTEP. For IPs, we have included age, sex, and education level alongside the presented quotes, as this information would not compromise the confidentiality of the participants.

## Trustworthiness

We used Lincoln and Guba's (1989) four criteria: credibility, dependability, confirmability, and transferability to establish trustworthiness in this study, an accepted approach in the content analysis method [41,42]. To ensure credibility, we pilot-tested the study tool, used an interview guide to provide broad and consistent coverage of topics, and conducted in-depth interviews with providers of different ages, genders, and educational backgrounds to capture variation in perspectives. To enhance dependability, we provide a transparent account of the study design, data collection, and analysis, and the coding framework was iteratively discussed and refined among researchers. To strengthen confirmability, we included two groups of providers (IPs and NTEP-affiliated personnel) to enable triangulation of perspectives, maintained an audit trail documenting coding decisions and revisions to the coding framework, and shared codes with the RA involved in data collection to ensure transparency. For transferability, we provided a detailed description of the study participants, the context, and the research process so that readers can assess the applicability of the findings to other similar settings.

## Ethical considerations

Ethical approval for this study was obtained from the UNSW Human Research Ethics Committee (HC210258) and the LFWB Human Research Ethics Committee (IILDS/IEC/001). Informed verbal consent was obtained from all participants, in accordance with the procedures approved by both ethics' committees. Before data collection, interviewers verbally administered a standardized consent script and provided participants with the opportunity to ask questions. Upon confirming their understanding, participants' verbal consent was audio-recorded. To maintain confidentiality, consent recordings were securely stored and kept separate from all other study data.

## Results

A total of 23 IPs and 11 NTEP-affiliated personnel participated in the study [Table 1]. Among IPs, the median age was 43 years, and the median work experience was 15 years. The majority were male (87%) with varied education levels: 17% had secondary or below, 48% post-secondary (post-sec), and 35% higher education. NTEP-affiliated personnel comprised three state-level officials, six District Health Office members, and two ASHAs. Among NTEP-affiliated personnel, 82% held higher education qualifications, with the two community-level Accredited Social Health Activists (ASHAs) reporting secondary education.

### IP's roles in TB care

The study categorizes the roles of IPs in TB care into three distinct sections (Table 2): 1) Current roles, encompassing TB-related activities IPs are already performing; 2) Potential roles, involving activities IPs are willing to adopt, or

**Table 1. Characteristics of study participants.**

| Characteristic | NTEP-Affiliated Personnel (n = 11) | Informal Healthcare Providers (n = 23) |
|---|---|---|
| Sex | Male: 7 (64%), Female: 4 (36%) | Male: 20 (87%), Female: 3 (13%) |
| Age (years), median (range) | 45 (30–58) | 43 (27–65) |
| Education | Secondary level: 2 (18%), Higher education: 9 (82%) | Secondary or below: 4 (17%), Post-secondary (post-sec): 11 (48%), Higher education: 8 (35%) |
| Level of the health system | State level: 3 (27%), District level: 6 (55%), Community level: 2 (18%) | Not applicable |
| Work experience (years), median (range) | 14 (2–30) | 15 (3–40) |

PLOS Global Public
Health

Table 2. Summary of the overall study findings (main categories and subcategories of IPs' current and potential roles in TB care).

| S. N | Major categories of roles | Role's subcategories, as expressed by IPs | Role's subcategories, as expressed by NTEP-affiliated personnel |
|---|---|---|---|
| 1 | Current roles in TB care | 1. Passive case finding and referral | 1. Passive case finding and referral |
| | | 2. Accompanying presumptive people with TB to a health facility | 2. Treatment supporter |
| | | 3. Collection/transportation of sputum samples | |
| | | 4. Treatment supporter | |
| 2 | Potential roles in TB care | 1. Health promotion and education | 1. Health promotion and education |
| | | 2. Active case finding | 2. Passive case finding and referral |
| | | 3. Passive case finding and referral | 3. Collection/transportation of sputum samples |
| | | 4. Collection/transportation of sputum samples | 4. Contact tracing |
| | | 5. Treatment supporter | 5. Treatment supporter |
| | | 6. Counselling – During TB treatment | 6. Counselling – During TB treatment |
| 3 | Roles not suitable for IPs | | 1. Clinical evaluation of TB |
| | | | 2. Treatment initiation |

NTEP-affiliated personnel suggest could be formally assigned to IPs; and 3) Roles not suitable for IPs, as advised by either IPs and/or NTEP-affiliated personnel.

**1 Current roles.** IPs expressed that they were currently undertaking four roles in TB care: 1) Passive case findings and referral, 2) Accompanying presumptive people with TB to a health facility, 3) Sputum collection/transportation, and 4) Treatment supporter. NTEP-affiliated personnel reported that IPs were currently involved in two roles in TB care in the community: 1) Passive case finding and referral, and 2) Treatment supporter, as shown in Table 3.

For passive case finding and referral, all NTEP-affiliated personnel confirmed West Bengal's government recently formalized IPs' engagement via an official order. An NTEP-affiliated participant stated, *"an official order was issued [..] to engage IPs for referral of presumptive people with TB, and the strategy was discussed for proper implementation with all districts [...]."* This marks a novel state-level formal involvement of IPs in NTEP. Out of the 23 IPs enrolled in this study, 10 reported being actively engaged in this government program. For the purposes of this study, "engaged IPs" refers to those IPs who received formal training from the government and were referring presumptive people with TB using the provided paper-based referral slips.

**Passive case finding and referral.** Passive case finding and referral was identified as a key role among IPs by both engaged and non-engaged IPs. Engaged IPs utilized referral slips for presumptive people with TB, as illustrated by an IP who shared, *"we have been given a referral slip, so whenever we suspect someone of having TB, we write down the patient's name, phone number, and our details and send them to the King (pseudonym used) hospital with the referral slip."* (IP, 38 M, Post-Sec). Non-engaged IPs also provided referrals as part of their routine practice, exemplified by an IP's statement, *"nobody has asked me to refer patients. I am doing it as part of my regular practice. Patients I have referred have tested positive for TB."* (IP, 44 F, Post-Sec).

IPs perceived successful referrals as a means to enhance their credibility within the community. This sentiment was captured by an IP who stated, *"with a proper referral, the patient will believe that I have the skills and capacity to manage their health conditions. It will improve their trust, and they will visit me in future for other health problems."* (IP, 54 M, High Edu.). However, challenges in making referrals, particularly in tribal areas, were encountered, with one IP recounting a failed attempt to persuade a patient to visit a health center.

IPs often initiated treatment for common symptoms such as cough and referred cases to hospitals if no improvement was observed or if severe TB symptoms emerged upon follow-up. The choice of referral destinations varied, with a general preference for government facilities due to free services, though some patients preferred private hospitals.

**Table 3. Tabular representation of IPs' current roles in TB care, based on perspectives of IPs and NTEP-affiliated personnel (classified based on the WHO people-centered model of TB care).**

| Type of care | Roles in TB care (classified by functions) | IPs | NTEP-affiliated personnel |
|---|---|---|---|
| **Prevention** | Health promotion and education | | |
| Sub-total (Roles in prevention) | | 0 | 0 |
| **Detection** | Active case finding | | |
| | Passive case finding and referral | ✔ | ✔ |
| | Accompany presumptive people with TB to a health facility | ✔ | |
| | Collection/ transportation of sputum samples | ✔ | |
| | Contact tracing | | |
| Sub-total (Roles in detection) | | 3 | 1 |
| **Treatment and support** | Treatment supporter | ✔ | ✔ |
| | Prevention and detection of adverse events and comorbidities | | |
| | Counseling – During TB treatment | | |
| Sub-total (Roles in treatment and support) | | 1 | 1 |
| Grand-total (All roles) | | 4 | 2 |

NTEP-affiliated personnel acknowledged IPs' formal role in referrals and the associated incentive system, which offered 500 INR (appx. 6 USD) for successful TB referrals. As shared by one NTEP-affiliated participant, "*I am aware that IPs are asked by health administration of different districts to refer a case of presumptive TB. We have also clarified that if a presumptive case is diagnosed with TB, the referring person would be given an incentive of five hundred Indian rupees.*" (NTEP-affiliated participant). The responses to the incentive system were mixed among engaged IPs; some expressed dissatisfaction when patients failed to use the referral slip, while others appreciated the financial recognition for their referrals.

**Accompany presumptive people with TB to a health facility.** A few IPs reported accompanying patients to health facilities when they identified someone with symptoms of TB, though this role was uncommon. One IP explained, "*if the patients have no one to help them to get to a health facility, I accompany those patients.*" (IP, 39 F, High Edu.). Another IP hesitated to perform this task due to the significant time required for visits to health centers (IP, 32 M, High Edu.).

**Collection/transportation of sputum samples.** Some IPs reported collaboration with local health facilities for the collection and transportation of sputum samples. One IP recounted being supplied with sputum containers to distribute to presumptive people with TB, with formal health workers coordinating the subsequent process. Another IP detailed his active role: "*next day I collected sample from patients, the container was given by TB center. I transported the sample to Bhwani-ganj (pseudonym used) primary health center for testing*" (IP, 57 M, Post-Sec.). Conversely, concerns about public perception deterred another IP from participating, as he expressed, "*if I collect sputum sample and transport it to a hospital, people might think that I get a commission for the work as a practice of providing commission for such work (lab-related) exist in the local market.*" (IP, 39 M, Post-Sec.).

**Treatment supporter.** A number of IPs had served as treatment supporters for people with TB, a role undertaken through informal local collaboration rather than formal assignment. NTEP-affiliated personnel acknowledged this contribution, especially in remote areas lacking health infrastructure. For instance, NTEP-affiliated participant highlighted that, "*Bharkata is a very remote area, and we don't have sufficient health infrastructure there. Ramesh (pseudonym used) provides treatment to patients following the regimen as per our guidance*" (NTEP-affiliated participant). An IP recounted his approach to ensuring medication adherence, stating, "*I worked as treatment supporter for a few years in our nearby hospital at (...). I noticed that many patients were not taking medicines; they threw the tablet. But I stayed in front of them and did not move until they took the medication*" (IP, 65 M, Post Sec.). This involvement was viewed positively by some

NTEP-affiliated personnel, with one noting, "*IPs act as a treatment supporter in a few places in Birbhum district. IPs have a good relationship in their area, so when we assign responsibility to IPs, they are doing well*" (NTEP-affiliated participant).

**2 Potential roles.** IPs showed willingness to undertake six roles in TB care, including continuing three existing roles and incorporating three new ones: 1) health promotion and education, 2) active case finding, and 3) counselling during TB treatment. Notably, IPs were not willing to continue accompanying patients to health facilities in the future, with this view expressed specifically by IPs and attributed to time constraints and disruption to their daily work routines. Conversely, NTEP-affiliated personnel were eager to assign IPs six roles, with two being continuations of existing roles and four new ones: 1) health promotion and education, 2) collection and transportation of sputum samples, 3) contact tracing, and 4) counselling during TB treatment. There was a significant overlap in perspectives between IPs and NTEP-affiliated personnel, agreeing on five out of seven potential roles, mainly within the detection domain, as detailed in Table 4.

**Health promotion and education.** Both IPs and NTEP-affiliated personnel expressed that there is significant benefit in engaging IPs for health promotion and education regarding TB. NTEP-affiliated personnel highlighted the persistent stigma and misconceptions within communities. As one NTEP-affiliated participant noted, "*people in the community are scared to hear about TB. They don't want to share if they have symptoms related to TB. There is a misperception that TB is a hereditary disease and that they won't get it if they live a healthy lifestyle. There is a need to overcome these misconceptions from the community*" (NTEP-affiliated participant). IPs expressed eagerness to combat these myths, drawing parallels to their contributions during the COVID-19 pandemic. One IP recounted, "*during COVID-19, I counselled people to wear a face mask, maintain physical distancing, wash their hands frequently and visit a government hospital if they had any symptoms related to COVID-19. We can educate people similarly in TB care*" (IP, 65 M, Post Sec.).

**Active case finding.** Some IPs expressed a willingness to engage in active case finding through door-to-door visits to identify presumptive people with TB and link them to government health facilities. As shared by one respondent, "*IPs can go to community and screen people for symptoms related to TB. It will help to identify people with TB early and prevent further transmission*" (IP, 38 M, Post-Sec.). However, IPs noted the need for a structured work schedule and financial compensation for time spent

**Table 4. Tabular representation of IPs' potential roles in TB care, based on perspectives of IPs and NTEP-affiliated personnel (classified based on the WHO people-centered model of TB care).**

| Type of care | Roles in TB care (classified by functions) | IPs | NTEP-affiliated personnel |
|---|---|---|---|
| **Prevention** | Health promotion and education | ✓ | ✓ |
| Sub-total (Roles in prevention) | | 1 | 1 |
| **Detection** | Active case finding | ✓ | |
| | Passive case finding and referral | ✓ | ✓ |
| | Accompany presumptive people with TB to a health facility | X | |
| | Collection/ transportation of sputum samples | ✓ | ✓ |
| | Contact tracing | | ✓ |
| Sub-total (Roles in detection) | | 3 | 3 |
| **Treatment and support** | Treatment supporter | ✓ | ✓ |
| | Counselling – During TB treatment | ✓ | ✓ |
| Sub-total (Roles in treatment and support) | | 2 | 2 |
| Grand-total (All roles) | | 6 | 6 |

Colour legend:

▨ New roles
▨ Continuation of previous roles
▨ Roles not willing to undertake in future

away from their clinics: "*we (IPs) are happy to provide time, but it needs to be defined like one day, two days or three days (every week) as well as we need to be supported with some financial compensation*" (IP, 38 M, Post-Sec.). One IP described his practice model as conducive to active screening, combining clinic hours with community visits: "*the nature of my practice is different as I offer services in my clinic in the morning for 2-3 hours, and then I take my cycle and go around my area asking people if they have any health-related illnesses. I visit every house in my area regularly*" (IP, 32 M, High Edu.).

**Passive case finding and referral.** NTEP-affiliated personnel and the 10 currently engaged IPs agreed on the importance of continuing passive case finding and referral roles, especially in remote areas where IPs are often the first point of contact for common illnesses. One NTEP-affiliated participant emphasized, "*in remote areas, people first go to IPs for fever, cold, cough, and other general health problems. People seek their services and advice. So, it is important to engage them in the TB program*" (NTEP-affiliated participant). The 13 IPs who were not yet engaged expressed a willingness to participate formally in the NTEP but highlighted the necessity of structured training to improve their diagnostic accuracy, as one IP noted, "*currently, we suspect and refer more patients, but the number of confirmed people with TB is low. If we get training, we can identify patients more accurately, and the number of referrals will be few*" (IP, 36 M, Post Sec.).

**Collection and transportation of sputum samples.** NTEP-affiliated personnel reported potential in involving IPs in the collection and transportation of sputum samples, particularly for sick patients and those with mobility challenges. IPs expressed willingness to take on this role, provided that sputum containers were supplied by the government system. NTEP-affiliated personnel suggested that the current incentive system could be expanded to cover IPs, offering "*190 rupees plus travel expenses for sputum sample collection and transportation to the designated microscopic center for testing*" (NTEP-affiliated participant). Another NTEP-affiliated participant highlighted a successful pilot where IPs served as sample carriers, noting, "*IPs were engaged as patient sample carriers in Bhanumati (pseudonym used) area, which is a cluster of islands separated by a river. This project was completed successfully with IPs*" (NTEP-affiliated participant).

**Contact tracing.** NTEP-affiliated personnel mentioned that engaging IPs could strengthen the government's contact tracing program under the NTEP. One NTEP-affiliated participant emphasized the potential effectiveness of IPs in this role, stating, "*IPs can be a good human resource for contact tracing. If anyone is diagnosed with TB or Drug Resistant (DR)-TB, it is important to trace everyone who were in close contact with the person with TB. IPs live and work close to the community and know almost everyone in their locality. It would be easy to trace people with their engagement*" (NTEP-affiliated participant).

**Treatment supporter.** IPs expressed interest in serving as treatment supporters for patients in their community. One IP highlighted their commitment, saying, "*I strongly believe that we can convince people to complete their course of treatment with love and care. We can go to their house, or they can come to our chamber (clinic), but either way, we can maintain regular contact with them*" (IP, 38 M, Post-Sec.). Another IP shared, "*I can keep the medicine and ask patients (as they will be local) to come to my chamber to take medicine*" (IP, 27 F, Higher Edu.). However, challenges related to clinic hours and the feasibility of home visits were noted, "*we [IPs] see patients in our clinic daily. But it will be difficult to maintain if I need to provide medication to 3-4 patients visiting their homes.*" (IP, 36 M, Post-Sec.).

NTEP-affiliated personnel also showed support for IPs as treatment supporters, particularly for seriously ill and bedridden patients. One NTEP-affiliated participant outlined the potential for IPs' engagement, "*IPs could be engaged in this role as we need treatment supporters for a certain percentage of patients, specifically those who are seriously ill and bedridden. Those working in their domicile area, such as IPs, can be best utilized for this role*" (NTEP-affiliated participant). The possibility of formally engaging IPs soon in this role was acknowledged, with suggestions to extend the current incentive system for treatment supporters to include IPs, which is currently undertaken by government health workers such as ASHAs.

Further engagement opportunities for IPs included administering injections with appropriate training and supporting people with DR-TB requiring long-term treatment. One NTEP-affiliated participant mentioned, "*people with DR-TB need a long-term treatment which may be two years in some cases. So, a treatment supporter is needed for them, and it is better if that person is from the local community with better acceptance such as IP*" (NTEP-affiliated participant).

Both groups identified the necessity for structured training. NTEP-affiliated personnel stressed, "*IPs can be engaged as treatment supporters, but they need proper sensitization and training before their engagement*" (NTEP-affiliated participant).

**Counselling during TB treatment.** Most providers from both groups identified counseling during TB treatment as a crucial role that can be undertaken by IPs. They emphasized the importance of counseling on diet, avoiding harmful habits, and particularly on medication adherence and managing adverse drug reactions. An NTEP-affiliated participant noted, "*in the case of DR-TB, they suffer from symptoms such as gastritis and vomiting. Because of this, they stop taking medicines. It is essential to have ongoing counselling, and IPs could be a great resource to convince patients*" (NTEP-affiliated participant). The significance of IPs in this role was underscored by their community acceptance, with one IP stating, "*they trust us very much. Even after consulting a formal doctor at a hospital, they come to us and ask for our opinion and advice*" (IP, 52 M, High Edu.). Some IPs recounted patient hesitancy towards medication due to fear and misinformation, as illustrated by an IP's experience, "*recently I gave medicines to a patient, but he did not want to continue as he felt weak after taking it. After regular counselling, I was able to convince the patient*" (IP, 44 F, Post-Sec.).

IPs agreed on the need for proper training to enhance their counseling capabilities, "*currently, if any TB patient visits me for some other health problems, I (IP) casually ask if they are taking medication or not. Sometimes they ask me about TB treatment, and I feel I cannot provide the right advice. But if we receive training, we can counsel them appropriately on drug side effects and the importance of taking medicine regularly*" (IP, 39 M, Sec.).

**3  Roles not suitable for IPs.**  Some NTEP-affiliated personnel strongly opposed assigning certain TB care roles to IPs, citing the need for more advanced training and clinical knowledge. They specifically highlighted clinical evaluation and treatment initiation as unsuitable tasks for IPs. One NTEP-affiliated participant clarified, "*IPs should never be entrusted to clinically diagnose TB as they are not doctors. Rather their role should be focused on case findings and referrals based on the presenting symptoms*" (NTEP-affiliated participant). Another underscored that initiating TB treatment falls outside IPs' scope, "*IPs should never be assigned responsibility to initiate treatment for a TB patient*" (NTEP-affiliated participant).

IPs did not express disagreement with these views. They consistently reported referring patients for further evaluation upon screening an individual with presumptive symptoms of TB, "*when I confirm that symptoms are related to TB, I ask the patient to visit King (pseudonym used) Hospital. I do not treat them, as people with TB should be managed only by a doctor*" (IP, 39 M, Primary). This referral practice stemmed from IPs' recognition that TB treatment requires a qualified doctor's care and their awareness of free TB care services and medicines at government facilities. One IP emphasized the accessibility of free treatment, "*we already know that six- or nine-month treatment is available for free from the government health center, that is why we say to patients that the service and medicine are free, and they just have to arrange transport*" (IP, 30 M, Post Sec.). Another IP mentioned, "*I advise patients to go to a government health center as tests and treatment are freely available there. I don't keep TB medicines in my clinic*" (IP, 38 M, Post Sec.).

## Discussion

This study was the first to explore IPs' current and potential roles in TB care in India, including both IPs and NTEP-affiliated personnel. We found that IPs were currently undertaking four roles in TB care, two of which were also confirmed by NTEP-affiliated personnel. Most identified roles were performed by IPs informally, either as part of their regular practice or in collaboration with local health facilities. There was alignment between the two groups of providers on the majority of potential roles (5 out of 7) for IPs in TB care. NTEP-affiliated personnel reported that two roles, clinical diagnosis of TB and initiation of treatment for people with TB, should not be assigned to IPs, citing the requirement for a high level of training and TB care knowledge.

In a scoping review previously undertaken by our team, we identified six roles assigned to IPs in TB care in LMICs [10]. When the roles of IPs were discussed with both IPs and NTEP-affiliated personnel in this study, IPs reported performing four roles, two of which were also acknowledged by NTEP-affiliated personnel. The two roles acknowledged by both groups

in this research, which were also identified in the scoping review for UAPs, were: 1) Passive case finding and referral, and 2) treatment supporter. These roles are prioritized in India's NSP under the Detect-Treat-Prevent-Build (DTBP) strategy [4]. Importantly, all roles identified in the scoping review and explored in this qualitative research can be carried out by non-medical professionals with proper training, according to the WHO's International Guideline on TB Care [43]. Moreover, all studies included in the scoping review reported a positive impact of IPs' involvement in enhancing TB care outcomes and services, a finding corroborated by multiple studies conducted among IPs in TB care [18–20,44,45]. These insights suggest that with appropriate training and support, IPs can be engaged in various roles in TB care within the framework of NTEP.

Among the three broader domains, as per the WHO people-centred model of TB care, detection was the area most emphasized by IP and NTEP-affiliated personnel for both current and potential roles. This emphasis aligns with the findings from our scoping review, where detection emerged as the most frequently assigned role to IPs in more than two-thirds (10 out of 13) of the studies included [10]. This focus is critical, as it aligns with India's NSP for TB Elimination, which advocates for the timely identification of presumptive people with TB [4]. Nonetheless, it could be advantageous for the NTEP to explore IPs' roles in health education, promotion, and counselling during treatment. This consideration stems from IPs' expressed willingness to undertake these roles and their potential to counteract prevalent stigma and misconceptions, as evidenced by their contribution during the COVID-19 pandemic [4,46]. Moreover, as India grapples with a significant burden of latent TB infections, the Government of India aims to expand TB preventive treatment to all household contacts of people with pulmonary TB and other high-risk groups, as recommended by the United Nations High-Level Meeting on TB [47,48]. Engaging IPs in roles such as latent TB screening could, therefore, benefit the NTEP.

In 2020, we conducted a quantitative TB care practice survey among 203 IPs at the same study site, documenting their approach to TB case management [17]. This practice study was focused primarily on the role of passive case finding and referral. However, the study's quantitative nature limited the in-depth exploration into actual practices, such as the reasoning and processes behind referrals and case management. Initially, the practice study identified instances of delayed referrals by IPs, which, according to this qualitative study, could be due to IPs' limited knowledge in screening presumptive people with TB. This observation is consistent with another knowledge survey carried out by our team, underscoring the need for further training in this area [26]. Moreover, this qualitative research also found that IPs typically treated patients for a set period, generally five to seven days, for any disease condition, potentially contributing to delays in referrals among IPs.

The previous practice survey also revealed that IPs prefer to refer presumptive or confirmed people with TB rather than provide treatment. This approach contrasts with previous research in India, which has documented instances of IPs administering treatment to people with TB drugs [15,16]. Our qualitative study observed no cases of IPs treating people with TB directly. Upon investigating this discrepancy, we learned that IPs' preference for referral stems from their awareness of the availability of free TB care services at government health facilities and their recognition that qualified medical practitioners should oversee such treatments. This preference for referral is also underpinned by IPs' belief in the value of proper referral practices, which, as noted in this study, are thought to foster patient trust and enhance the reputation of IPs within their communities. This finding marks a significant empirical contribution to the discourse on IPs' involvement in TB care, addressing the debated role of IPs in the direct treatment of people with TB [15,16,49].

The characteristics of IPs and their relationships within the community can significantly benefit the NTEP. The trust and connections IPs have in their communities could positively influence any TB care roles they undertake. Furthermore, IPs' in-depth knowledge of their communities positions them well for specific tasks, such as active case finding and contact tracing. Their recognized and established roles as treatment providers in their communities offers them a unique opportunity to refer presumptive people with TB. Their closeness to the community makes them suitable to act as treatment supporters. The value of these characteristics was affirmed by NTEP-affiliated personnel in this study and is corroborated by prior research [50,51].

In addition to these positive attributes, the willingness expressed by IPs to become formally involved (for those not currently participating) in the NTEP and to expand their roles (for those currently engaged) represents an opportunity for the NTEP to utilize this large cadre of the health workforce. A similar willingness to work closely with the formal system among

IPs was discovered in a previous quantitative study conducted by our team with 203 IPs, as well as in other similar studies focusing on IPs [17,52]. NTEP-affiliated personnel also showed enthusiasm, demonstrated by their willingness to assign additional TB care roles to IPs within the NTEP and appreciation for their current contributions. However, to successfully engage IPs in the NTEP, it is crucial to consider the critical perspectives expressed in this study. These include IPs' preference for certain TB care roles, their need to balance time spent in their clinics as primary care providers, the necessity for training, and the provision of incentives.

## Limitations

There are several limitations to this study. Firstly, due to the COVID-19 situation in India, most interviews were conducted remotely, which might have restricted our ability to gather in-depth information compared to in-person interviews. Secondly, this study focused on IPs who primarily practice allopathic medicine (UAPs), limiting the applicability of the findings to other types of IPs, such as traditional healers and traditional birth attendants. Third, the study was limited to the views of NTEP-affiliated personnel up to the state level and did not include formal healthcare providers such as physicians, whose perspectives may differ, particularly with service delivery roles. Central-level TB officials involved in policy and guideline formulation were not included due to difficulties accessing these individuals, due to the ongoing COVID-19 pandemic during data collection. Lastly, all the NTEP-affiliated personnel included in the study were affiliated with the government health system, so the study findings may not reflect the perspectives of the healthcare providers working in the private sector, such as private physicians.

## Conclusion

This research provides crucial insights into the roles of IPs in TB care in India, marking a significant step toward understanding their involvement and potential within the NTEP. NTEP-affiliated personnel acknowledged two roles currently carried out by IPs, but IPs themselves reported undertaking four distinct roles in TB care at the community level. There was a notable consensus on the majority of potential roles identified by both groups, indicating a broad opportunity for integrating IPs into the NTEP framework. Despite their current marginalization in India's national TB policies and programs, this study qualitatively captures the role and contributions of IPs in TB care. It also offers guidance by outlining potential roles for the future engagement of IPs in the NTEP.

## Supporting information

**S1 Text. Categorization Matrix.**
(PDF)

**S2 Text. Interview Guide.**
(PDF)

**S1 File. COREQ Checklist.**
(PDF)

**S1 Checklist. Inclusivity in global research.**
(DOCX)

## Acknowledgments

We extend our gratitude to the State TB cell of West Bengal and the District TB cell of Birbhum District for their invaluable support during the study's implementation. Our deepest appreciation goes to all the IPs who participated in this study and shared their insights and experiences. Additionally, the PhD student (PT) would like to thank UNSW Sydney for the support provided through the Scientia PhD scholarship.

## Author contributions

**Conceptualization:** Poshan Thapa, Padmanesan Narasimhan, John J. Hall, Rohan Jayasuriya, Partha Sarathi Mukherjee, Kristen Beek.

**Data curation:** Poshan Thapa, Partha Sarathi Mukherjee, Dipesh Kr Das, Kristen Beek.

**Formal analysis:** Poshan Thapa, Dipesh Kr Das, Kristen Beek.

**Methodology:** Poshan Thapa, Padmanesan Narasimhan, John J. Hall, Rohan Jayasuriya, Partha Sarathi Mukherjee, Kristen Beek.

**Project administration:** Poshan Thapa, Padmanesan Narasimhan, John J. Hall, Rohan Jayasuriya, Partha Sarathi Mukherjee, Dipesh Kr Das, Kristen Beek.

**Software:** Poshan Thapa, Dipesh Kr Das, Kristen Beek.

**Supervision:** Padmanesan Narasimhan, John J. Hall, Rohan Jayasuriya, Partha Sarathi Mukherjee, Kristen Beek.

**Validation:** Padmanesan Narasimhan, John J. Hall, Rohan Jayasuriya, Partha Sarathi Mukherjee, Dipesh Kr Das, Kristen Beek.

**Writing – original draft:** Poshan Thapa, Kristen Beek.

**Writing – review & editing:** Poshan Thapa, Padmanesan Narasimhan, John J. Hall, Rohan Jayasuriya, Partha Sarathi Mukherjee, Dipesh Kr Das, Kristen Beek.

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
