## [Decision Letter · Decision Letter 0]

19 Feb 2025

PGPH-D-24-02750

Exploring current and potential roles of informal healthcare providers in tuberculosis care in West Bengal, India: a qualitative content analysis

Dear Dr. Thapa,

Thank you for submitting your manuscript to PLOS Global Public Health. After careful consideration, we feel that it has merit but does not fully meet PLOS Global Public Health’s publication criteria as it currently stands. Therefore, we invite you to submit a revised version of the manuscript that addresses the points raised during the review process.

Please note that we have only been able to secure a single reviewer to assess your manuscript. Please see the reviewer's comments below. We are issuing a decision on your manuscript at this point to prevent further delays in the evaluation of your manuscript. Please be aware that the editor who handles your revised manuscript might find it necessary to invite additional reviewers to assess this work once the revised manuscript is submitted. However, we will aim to proceed on the basis of this single review if possible. 

Could you please carefully revise the manuscript to address all comments raised?

We look forward to receiving your revised manuscript.

Kind regards,

Steve Zimmerman, PhD

PLOS Staff Editor

Journal Requirements:

1. In the ethics statement in the Methods, you have specified that verbal consent was obtained. Please provide additional details regarding how this consent was documented and witnessed, and state whether this was approved by the IRB

2. Please include a complete copy of PLOS’ questionnaire on inclusivity in global research in your revised manuscript. Our policy for research in this area aims to improve transparency in the reporting of research performed outside of researchers’ own country or community. The policy applies to researchers who have travelled to a different country to conduct research, research with Indigenous populations or their lands, and research on cultural artefacts. The questionnaire can also be requested at the journal’s discretion for any other submissions, even if these conditions are not met.  Please find more information on the policy and a link to download a blank copy of the questionnaire here: https://journals.plos.org/globalpublichealth/s/best-practices-in-research-reporting. Please upload a completed version of your questionnaire as Supporting Information when you resubmit your manuscript.

Additional Editor Comments (if provided):

Reviewers' comments:

Reviewer's Responses to Questions

**Comments to the Author**

1. Does this manuscript meet PLOS Global Public Health’s publication criteria?

Reviewer #1: Yes

2. Has the statistical analysis been performed appropriately and rigorously?

Reviewer #1: N/A

3. Have the authors made all data underlying the findings in their manuscript fully available (please refer to the Data Availability Statement at the start of the manuscript PDF file)?

Reviewer #1: No

4. Is the manuscript presented in an intelligible fashion and written in standard English?

Reviewer #1: Yes

Reviewer #1: The work presented here explores current and potential roles of a subset of informal providers – Untrained Allopathic Practitioners (UAPs) – in TB care in West Bengal in India. Overall, the research employed sound methodology by drawing on perspectives of both informal and formal providers to identify these roles. In particular, it is thoughtful to include formal providers because UAPs sometimes share a close relationship with the formal providers (particularly ASHA in this case). From a methodological perspective, use of the content analysis methodology outlined by Elo and Kygnas, drawing on previous scoping review and people-centered care framework to identify potential roles, examining potential roles and expanding the roles’ framework, and presentation of the results in a tabular format for programmatic guidance is well appreciated.

I suggest considering following revisions.

1. The results tables 2 and 3 identify the role as “Accompany suspected TB cases to a health facility”, whereas the associated text says presumptive. In line with use of respectful language, I suggest avoiding use of “suspect”. Please refer to the language guide: https://www.stoptb.org/words-matter-language-guide

2. The focus is on UAPs. Although it is defined early-on, I disagree with the use of IP in lieu of UAP. The introduction mentions that only UAP subset of IP was involved and that should come across the entire paper and abstract. Using UAP instead of IP can be misleading by increasing the scope of interpretation to the different types of IP.

3. On a related note, the authors decision to involve various levels of formal providers has improved the rigor of this work. However, formal providers involved were “three state-level officials, six District Health Office members, and two ASHA.” I would suggest authors to use another term instead of FP because typically FPs mean trained clinicians. The above set of people will be NTEP officials and community health workers. It is an overreach to call them FP, which should have included physician, other health workers, and potentially street-level NTEP staff. I note that authors do mention some additional details in limitation but that is insufficient.

4. As per table 3 on potential role, UAPs do not wish to accompany presumptive people to the facility. It would be useful to understand the underlying details in the accompanying text. Was this response universal or a one-off? Whom did it come from – UAP or a certain FP cadre? What were their reasons?

5. As the introduction clearly explains, the policies for IP vary between states. It would be helpful to contextualize the NTEP’s approach. For instance, does NTEP have an approach towards engaging them? Are there any incentives – financial or other – for their engagement? Are they part of public-part mix or similar programs? Can they notify people with TB directly to the NTEP?

**Do you want your identity to be public for this peer review?** For information about this choice, including consent withdrawal, please see our Privacy Policy

Reviewer #1: **Yes: ** Tushar Garg

---

## [Decision Letter · Decision Letter 1]

26 Jun 2025

PGPH-D-24-02750R1

Exploring current and potential roles of informal healthcare providers in tuberculosis care in West Bengal, India: a qualitative content analysis

Dear Dr. Thapa,

Thank you for submitting your manuscript to PLOS Global Public Health. After careful consideration, we feel that it has merit but does not fully meet PLOS Global Public Health’s publication criteria as it currently stands. Therefore, we invite you to submit a revised version of the manuscript that addresses the points raised during the review process.

The manuscript has been evaluated by two reviewers, and their comments are available below.

The reviewers have raised a number of concerns that need attention. They request additional information on methodological aspects of the study such as clarification on the type of qualitative design utilized in the study, how participants were recruited; and they request you report the participants’ characteristics broadly and provide a table of key characteristics such as age, sex, education and work experience, in addition to providing a table which presents the main study findings.

Could you please revise the manuscript to carefully address the concerns raised?

We look forward to receiving your revised manuscript.

Kind regards,

Katrien G. Janin, PhD

Staff Editor

Reviewers' comments:

Reviewer's Responses to Questions

**Comments to the Author**

Reviewer #1: All comments have been addressed

Reviewer #2: (No Response)

publication criteria?

Reviewer #1: Yes

Reviewer #2: Partly

3. Has the statistical analysis been performed appropriately and rigorously?

Reviewer #1: N/A

Reviewer #2: N/A

4. Have the authors made all data underlying the findings in their manuscript fully available (please refer to the Data Availability Statement at the start of the manuscript PDF file)?

Reviewer #1: No

Reviewer #2: Yes

5. Is the manuscript presented in an intelligible fashion and written in standard English?

Reviewer #1: Yes

Reviewer #2: Yes

Reviewer #1: Thanks for considering the comments. The current version reads very well after the exhaustive and careful revision. I have no further comments.

Reviewer #2: This qualitative study sought to explore the perspectives of informal healthcare providers (IPs) and Formal Providers (FPs) regarding IPs' current and potential roles in TB care in West Bengal, India.

Although the study's purpose has merit, the manuscript must be written as a clear and concise independent paper. In many sections, it read like the residual results of a larger study. The following issues should be addressed;

Materials and Methods

• Line 126: Specify the type of qualitative design utilised in the study. Line 178 suggests it was exploratory in nature.

• 165: Although the previous study was cited and purposive sampling was mentioned, briefly describe how participants were recruited into the main study. How were prospective participants identified, approached and recruited? How many participants refused to participate in the study? If any, what rationale did those who refused to participate in the qualitative study give?

• Specify the location within the district from which participants were sampled. Eg. In health facilities, within communities?

• 180: Although participant availability is mentioned as a requirement for participation, specify the inclusion and exclusion criteria for both populations/groups of participants ie (IPs and NTEP-personnel).

• Indicate the number of interviews conducted via phone/face to face.

• Describe how trustworthiness was achieved in this qualitative study.

Results

• 262: Reporting descriptive statistics (such as median age, percentage being majority etc) may be problematic in a qualitative study, as this. Consider reporting the participants’ characteristics broadly and provide a table of key characteristics such as age, sex, education and work experience.

• Provide a table which presents the main study findings (headings: 1, 1.1, 1.1.1) to orient readers.

• 290: Table 2 is unclear. The representation of a related scoping review's findings in this paper, together with the results of this qualitative paper is challenging, especially for qualitative research fundamentalists or purists. It may be useful to discuss how the results of previous papers relate to the findings in the discussion section.

• 307: “Post-Sec” should probably be explained when used the first time or in the methods.

• 320 – 323: It may be helpful to attribute quotes to participants with ID numbers, eg, IP 1, IP2, or IP23.

**Do you want your identity to be public for this peer review?** For information about this choice, including consent withdrawal, please see our Privacy Policy

Reviewer #1: **Yes: ** Tushar Garg, Stop TB Partnership

Reviewer #2: No

---

## [Decision Letter · Decision Letter 2]

20 Oct 2025

Exploring current and potential roles of informal healthcare providers in tuberculosis care in West Bengal, India: a qualitative content analysis

PGPH-D-24-02750R2

Dear Dr. Thapa,

We are pleased to inform you that your manuscript 'Exploring current and potential roles of informal healthcare providers in tuberculosis care in West Bengal, India: a qualitative content analysis' has been provisionally accepted for publication in PLOS Global Public Health.

Best regards,

Julia Robinson

Executive Editor

Reviewer Comments (if any, and for reference):

Reviewer's Responses to Questions

**Comments to the Author**

Reviewer #1: All comments have been addressed

Reviewer #2: All comments have been addressed

publication criteria?

Reviewer #1: Yes

Reviewer #2: Yes

3. Has the statistical analysis been performed appropriately and rigorously?

Reviewer #1: Yes

Reviewer #2: Yes

4. Have the authors made all data underlying the findings in their manuscript fully available (please refer to the Data Availability Statement at the start of the manuscript PDF file)?

Reviewer #1: No

Reviewer #2: Yes

5. Is the manuscript presented in an intelligible fashion and written in standard English?

Reviewer #1: Yes

Reviewer #2: Yes

Reviewer #1: Authors have comprehensively revised the manuscript to address all the comments of the other reviewer. I recommend accepting the manuscript for publication.

Reviewer #2: The authors have addressed the key issues raised. Congratulations on this insightful study.

**Do you want your identity to be public for this peer review?** For information about this choice, including consent withdrawal, please see our Privacy Policy

Reviewer #1: **Yes: ** Tushar Garg, Stop TB Partnership

Reviewer #2: No
